

# Automatic visual recognition for leaf disease based on enhanced attention mechanism

Yumeng Yao[1], Xiaodun Deng[1], Xu Zhang[2], Junming Li[1], Wenxuan Sun[1] and Gechao Zhang[3]

[1] School of Engineering, Xi'an International University, Xi'an, China
[2] Qilu University of Technology (Shandong Academy of Sciences), Jinan, China
[3] School of Chemistry, Xi'an Jiaotong University, Xi'an, China

## ABSTRACT

Recognition methods have made significant strides across various domains, such as image classification, automatic segmentation, and autonomous driving. Efficient identification of leaf diseases through visual recognition is critical for mitigating economic losses. However, recognizing leaf diseases is challenging due to complex backgrounds and environmental factors. These challenges often result in confusion between lesions and backgrounds, limiting information extraction from small lesion targets. To tackle these challenges, this article proposes a visual leaf disease identification method based on an enhanced attention mechanism. By integrating multi-head attention mechanisms, this method accurately identifies small targets of tomato lesions and demonstrates robustness in complex conditions, such as varying illumination. Additionally, the method incorporates Focaler-SIoU to enhance learning capabilities for challenging classification samples. Experimental results showcase that the proposed algorithm enhances average detection accuracy by 10.3% compared to the baseline model, while maintaining a balanced identification speed. This method facilitates rapid and precise identification of tomato diseases, offering a valuable tool for disease prevention and economic loss reduction.

## INTRODUCTION

Visual recognition plays a pivotal role in various domains (*Tian et al., 2023*; *Yao et al., 2021*) such as smart production, autonomous driving, and intelligent perception. In the context of smart production, automatic leaf disease recognition can significantly mitigate agricultural economic losses (*Martinez, 2007*; *Ananthi & Varthini, 2012*). Many crop diseases originate from the leaves and subsequently affect the entire plant, leading to a decline in crop yield and quality (*Yağ & Altan, 2022*; *Karasu & Altan, 2022*). Therefore, timely and accurate identification of leaf disease types is crucial for early detection and diagnosis of tomato diseases (*Zhang, Shang & Wang, 2015*). Despite notable advancements in automatic recognition, existing methods encounter challenges due to the complexity of leaf textures, similarity in disease leaf appearances, and environmental factors (*Kaur, Pandey & Goel, 2018*; *Al-Hiary et al., 2011*; *Zhou et al., 2021*). Traditional machine learning-based algorithms struggle to extract features from small disease areas and lack

Corresponding authors
Yumeng Yao,
yumeng_yao2023@163.com
Gechao Zhang,
zhanggc100@163.com

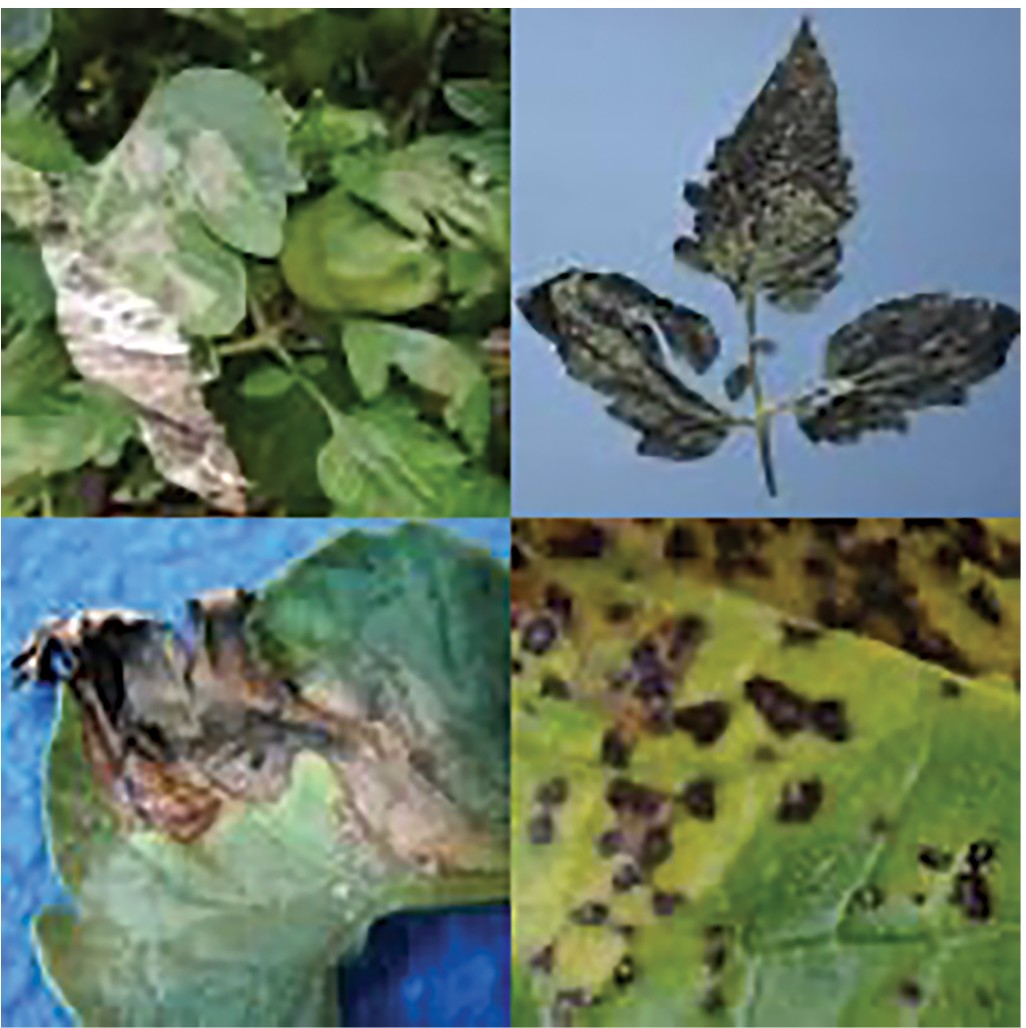

**Figure 1 Tomato leaf disease display.** Image source credit: PlantDoc dataset.

adaptability (*Al Bashish, Braik & Bani-Ahmad, 2011*; *Vaishnnave et al., 2019*; *Rumpf et al., 2010*; *Shruthi, Nagaveni & Raghavendra, 2019*), while deep learning approaches lack specialized methods for extracting features from tomato disease leaves (*Shrivastava & Pradhan, 2021*; *Arivazhagan et al., 2013*; *Patil & Kumar, 2017*).

Symptoms of diseases manifesting on crop leaves (*Kaur, Pandey & Goel, 2018*; *Al-Hiary et al., 2011*; *Zhou et al., 2021*) can be accurately identified using vision-based methods for crop disease recognition, which involve extracting disease-specific features from images of crop leaves. However, the diverse spectrum of tomato diseases poses a significant challenge in distinguishing diseased leaves based solely on texture, shape, and color. The presence of complex backgrounds and the small size of disease spots further exacerbate this challenge, complicating the differentiation of diseased areas from healthy foliage. Moreover, variations in environmental conditions, such as lighting, introduce additional complexities to the recognition of tomato diseases, as depicted in Fig. 1. These multifaceted challenges

underscore the critical focus of ongoing research efforts among experts and scholars in the field of tomato disease recognition.

Traditional algorithms for crop disease analysis (*Al Bashish, Braik & Bani-Ahmad, 2011*; *Shruthi, Nagaveni & Raghavendra, 2019*; *Vaishnnave et al., 2019*) typically rely on analyzing color, texture, shape, and other features in images of crop disease leaves to classify disease types and localize affected areas. However, these methods often face challenges in accurately extracting features from small disease-affected regions and may fail to capture the dynamic changes in tomato crop diseases influenced by variables such as growth stage, geographic region, and environmental conditions. Furthermore, traditional approaches do not embrace the end-to-end feature learning paradigm, necessitating iterative processes of feature screening, selection, and evaluation. This lack of integration can lead to time-consuming workflows and limits the applicability of these methods across diverse agricultural settings.

An alternative approach to tomato leaf disease recognition utilizes deep learning, which emulates the neural network architecture and cognitive processes of the human brain, enabling autonomous perception and extraction of intrinsic characteristics related to tomato leaf diseases. However, many current deep learning-based methods for crop disease recognition rely on general-purpose frameworks that lack specialized techniques for effectively extracting visual features from tomato disease leaves. As a result, when confronted with complex backgrounds and varying lighting conditions, these models encounter challenges in accurately distinguishing between lesions and the background, thereby compromising the extraction of features from small, targeted lesions.

To address these challenges, this article introduces an enhanced attention-based algorithm for tomato disease identification. Firstly, the algorithm achieves a balance between speed and precision by leveraging YOLOv4 (tiny) as the base model (*Bochkovskiy, Wang & Liao, 2020*), enabling automatic learning and extraction of image features related to tomato diseases across diverse samples. This adaptability facilitates effective handling of dynamic changes in disease characteristics. Secondly, the enhanced attention module integrates multi-head attention mechanisms to enhance feature learning and characterization capabilities of the deep learning model, focusing on scale-awareness, spatial awareness, and task awareness. This enhancement significantly improves the extraction of critical features from small targets within tomato lesions. Furthermore, the algorithm incorporates Focaler-SIoU (*Zhang & Zhang, 2024*) to address a large number of challenging samples during training. This adaptation allows the recognition algorithm to prioritize attention on difficult samples, thereby further enhancing the accuracy of tomato leaf disease identification. The main contributions of this article can be summarized as follows:

1) Development of a recognition algorithm that effectively balances speed and precision for tomato disease identification.
2) Proposal of a tomato disease identification algorithm integrating an enhanced attention module to autonomously learn and extract crucial features from tomato lesions. This approach mitigates challenges related to feature extraction from tomato lesions,

effectively suppressing irrelevant information such as background noise and illumination variations.

3) Extensive experiments conducted on real-world datasets demonstrate the efficacy of the proposed algorithm. The experimental results showcase superior disease recognition precision compared to general detection models, while maintaining a well-balanced recognition speed.

## Related work

### Methods of crop disease identification

In recent years, crop disease identification methods have undergone significant evolution, largely due to advancements in object recognition detection algorithms and the integration of deep learning techniques. These methods have found successful applications across various domains, including medical image recognition (*Huang et al., 2021*), pedestrian gait recognition (*Dou et al., 2022*), and crop disease recognition (*Verma et al., 2021*). Deep learning, functioning as an end-to-end target recognition and detection algorithm, mimics the network structure and operational mechanisms of the human brain, enabling independent perception and learning of intrinsic features of the target. Several classical algorithms, such as Single Shot MultiBox Detector (SSD) (*Liu et al., 2016*), Region-based Convolutional Neural Network (R-CNN) (*Girshick et al., 2014*), Spatial Pyramid Pooling Network (SPP-Net) (*He et al., 2015*), Region-based Fully Convolutional Network (R-FCN) (*Dai et al., 2016*), and You Only Look Once (YOLO) (*Redmon et al., 2016*), have been developed based on deep learning.

Among these algorithms, YOLO stands out for its transformation of the detection problem into a regression problem and its ability to achieve real-time target detection using a convolutional neural network. Consequently, an increasing number of researchers are integrating deep learning with crop disease recognition (*Xinming & Hong, 2023*; *Xue et al., 2023*; *Morbekar, Parihar & Jadhav, 2020*).

For instance, *Mohanty, Hughes & Salathé (2016)* trained a deep convolutional neural network model using a plant leaf disease dataset, achieving recognition of 26 crop diseases. Notably, the model also recognized disease images not present in the training set. *Too et al. (2019)* explored the application of a fine-tuned deep learning model for plant disease classification using the PlantVillage dataset, with experimental results demonstrating that the DenseNet network model achieved the best recognition performance.

*Turkoglu, Hanbay & Sengur (2022)* proposed a model for detecting apple pests and diseases based on a multi-model LSTM convolutional neural network. This hybrid model combined the LSTM network with a pre-trained CNN model, with experimental results showing higher accuracy compared to other models. *Nachtigall, Araujo & Nachtigall (2016)* introduced a classification method for apple tree diseases based on convolutional neural networks. By leveraging CNN, relevant features of apple tree diseases were learned from the data, with experimental results indicating that the trained convolutional neural network outperformed human experts in identifying apple tree diseases.

In 2021, *Shill & Rahman (2021)* developed an accurate plant disease detection system using YOLOv3 (*Redmon & Farhadi, 2018*) and YOLOv4 (*Bochkovskiy, Wang & Liao, 2020*), respectively, achieving the detection of diseases related to 17 plant leaves. Experimental results demonstrated the system's high accuracy and applicability. *Ganesan & Chinnappan (2022)* proposed a rice disease recognition algorithm based on a mixed deep learning model, in which a YOLO classifier replaced the fully connected layer of the ResNet model. Experimental results showed that the recognition algorithm achieved high accuracy. The TC-MRSN model (*Wang et al., 2024*) excels in diagnosing maize leaf diseases under complex conditions by employing a dual-branch system to effectively capture texture and color features with high precision. The SENet approach (*Wen et al., 2024*) integrates a multi-scale residual network with Squeeze-and-Excitation mechanisms for precise recognition of mulberry leaf diseases. A method for recognizing pepper leaf diseases (*Fu, Guo & Huang, 2024*) introduces a lightweight CNN model based on the GGM-VGG16 architecture. Trained on images against a human palm background, it operates as a mobile application for efficient and accurate diagnosis in field conditions. The CoffeeNet model (*Nawaz et al., 2024*), employing a novel deep learning strategy, addresses the challenge of accurately diagnosing coffee plant leaf diseases. It incorporates spatial-channel attention mechanisms within a ResNet-50-based architecture and utilizes the CenterNet framework for streamlined one-step detection. Furthermore, the GhostNet Triplet YOLOv8s algorithm (*Li et al., 2024*) enhances maize leaf disease detection, providing a more efficient and accurate solution for real-time agricultural diagnostics. Additionally, an approach (*Deari & Ulukaya, 2024*) combines Inception v3 for classification with YOLOv5x for precise symptom localization, enhancing early detection and preserving yield.

Although the above methods have made significant progress in terms of recognition accuracy, the reality of plant disease detection requires high detection efficiency under varying light and background conditions. We will enhance the ability to detect plant diseases in these challenging environments.

### Identification method of tomato disease

*Durmuş, Güneş & Kırcı (2017)* utilized deep learning for tomato leaf disease detection using the PlantVillage dataset, conducting tests on the AlexNet and SqueezeNet deep learning network models. Experimental results highlighted SqueezeNet's lightweight nature, enabling real-time identification of nine tomato diseases. *Brahimi, Boukhalfa & Moussaoui (2017)* proposed a tomato disease recognition algorithm based on CNN, leveraging CNN's automatic feature extraction to visualize disease regions in tomato leaves, achieving high accuracy as indicated by experimental results. *Rangarajan, Purushothaman & Ramesh (2018)* introduced a tomato disease classification algorithm based on pre-trained deep learning network models, specifically AlexNet and VGG16-net, analyzing the effects of image quantity, small batch size weight, and bias learning rate on tomato disease recognition performance.

While these algorithms applied deep convolutional neural networks to tomato leaf disease recognition using the PlantVillage dataset, it is crucial to note that these datasets

lack characteristics such as complex backgrounds, variable illumination, and close contrast between disease spots and the background, as illustrated in Fig. 1. Therefore, when applying these deep learning network models to datasets similar to the one shown in Fig. 1 for identifying tomato leaf diseases, detection accuracy is often compromised. However, detection speed can be higher, making it more feasible to achieve the desired recognition effect.

*Fuentes et al. (2017)* proposed a recognition method for tomato diseases and insect pests based on deep learning, exhibiting robustness in complex environmental conditions and enabling effective recognition of nine tomato diseases, thereby facilitating early prevention of tomato diseases and insect pests. *Mohandas, Anjali & Varma (2021)* introduced a real-time plant leaf disease recognition algorithm based on YOLOv4 (tiny), capable of recognizing various crop diseases, including those affecting tomatoes, mangoes, strawberries, beans, and potatoes. Experimental results demonstrated that the proposed algorithm can achieve early-stage recognition of plant diseases. *Lin et al. (2017)* proposed a tomato and apple leaf disease recognition algorithm based on YOLOv4, utilizing EPC to optimize the algorithm's learning rate, ultimately achieving recognition of eight tomato diseases and insect pests. Additionally, due to its ability to focus on specific regions akin to human vision, attention mechanisms have gradually found applications across various domains, yielding fruitful outcomes (*Liu et al., 2022*). Nevertheless, there is scarce research leveraging attention mechanisms in conjunction with YOLO for leaf disease classification.

## METHODOLOGY

In the context of tomato disease recognition, we treated tomato leaf disease recognition as a small target recognition problem, with a specific focus on tomato disease lesions. To tackle this, we proposed a recognition algorithm that combines multi-head attention mechanisms and focuses more on hard samples, enabling more precise tomato disease recognition. The overall pipeline of our approach is illustrated in Fig. 2.

To meet the requirements of both speed and precision in disease identification, we selected YOLOv4 (tiny) as the base model. However, due to the presence of complex backgrounds and small lesion areas, we incorporated an enhanced attention module DyHead (*Dai et al., 2021*) for tomato disease features. By integrating multi-head attention mechanisms such as scale-aware, spatial-aware, and task-aware within the DyHead, we further enhanced the capability of capturing features related to tomato lesions, resulting in a significant improvement in the representation ability of the model.

Furthermore, due to the diverse types of tomato leaf diseases and variations in shape, color, and texture caused by individual differences, there are numerous hard samples encountered during the recognition process. To address this, we introduced a Focaler-SIoU method into our algorithm, which focuses more on recognizing hard samples of tomato leaf diseases.

In the following sections, we provide a detailed description of our proposed model architecture.

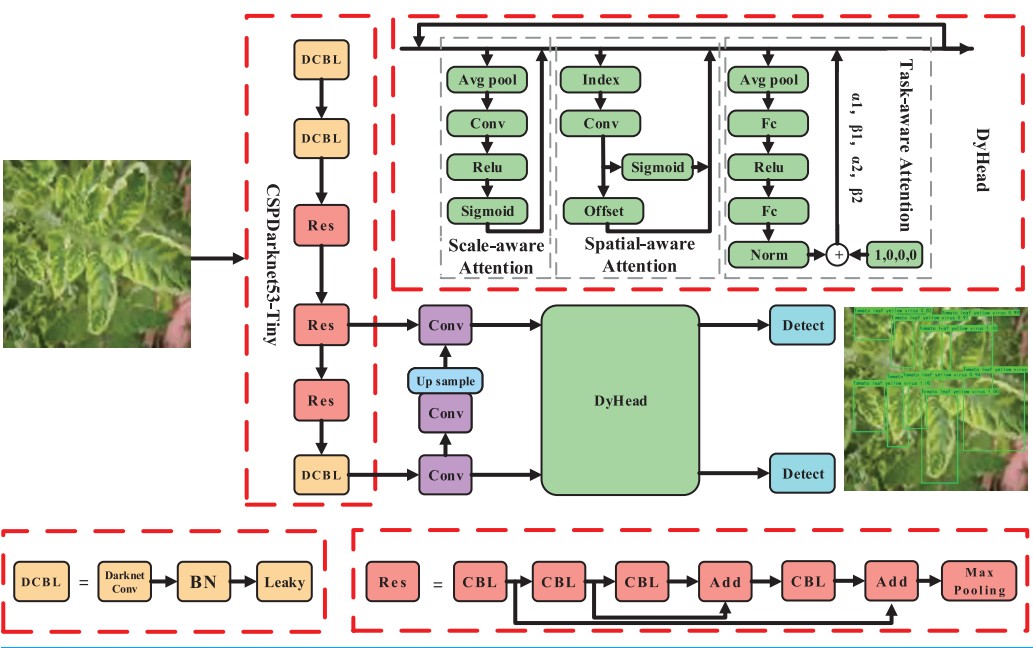

**Figure 2 The pipeline of the proposed method.** Image source credit: PlantDoc dataset.

## Enhanced module for detection head based on multi-head attention mechanisms

In the context of tomato crop diseases, various factors contribute to the dynamic changes in disease characteristics, such as growth stage, planting region, and climate conditions. Consequently, the manifestation of the same tomato leaf disease in images can vary significantly. Additionally, the background of tomato disease leaves is often complex, and the small target characteristics of disease spots may not be obvious. This makes it challenging to distinguish disease spots from the background, and the recognition process can be influenced by environmental conditions, such as varying lighting.

Given these challenges, it becomes crucial to focus on effective feature extraction in the lesion area, considering factors like complex backgrounds and small lesion areas. Inspired by the attention mechanism (*Hu, Shen & Sun, 2018*; *Woo et al., 2018*; *Wang et al., 2020*), which leverages the self-learning ability to determine the importance of features from a vast amount of information, we introduced an enhanced module, DyHead (as shown in Fig. 2), that combines multi-head attention mechanisms. This module enhances the feature learning and representation ability of the detection head. It focuses more on the effective features of the small targets of tomato spots while suppressing irrelevant information, such as background noise and illumination changes. This approach aims to solve the challenge of extracting key features from small targets in tomato leaf disease recognition.

The DyHead module can be represented as follows:

$$W(F) = \alpha_C[\alpha_S[\alpha_L(F) \cdot F] \cdot F] \cdot F, \tag{1}$$

where $\alpha(\cdot)$ represents the attention function, and $F$ represents the feature. $\alpha_L(\cdot)$, $\alpha_S(\cdot)$, and $\alpha_C(\cdot)$ respectively represent the scale-aware attention, spatial-aware attention, and task-
aware attention. By combining multi-head attention mechanisms, the representation ability of the detection head is further enhanced.

**Scale-aware attention** $\alpha_L(F)$. Due to variations in growth stages and planting regions, which lead to differences in tomato leaf sizes, the scale-aware attention assists the model in adaptively perceiving targets of different sizes by fusing features from different scales:

$$\alpha_L(F) \cdot F = \sigma\left[f\left[\frac{1}{SC}\sum_{S,C}F\right]\right] \cdot F, \quad where \quad \sigma(X) = \max\left(0, \min\left(1, \frac{X+1}{2}\right)\right), \tag{2}$$

where $f(\cdot)$ represents the 1D convolution operation with a convolution kernel size of $1 \times 1$. $\sigma(X)$ is a hard sigmoid function.

**Spatial-aware attention** $\alpha_S(F)$. Due to the complexity of backgrounds on tomato disease leaves, recognizing tomato disease leaf targets from such complex backgrounds can be challenging. The spatial-aware attention can effectively perceive the relationship between tomato leaves at different spatial positions:

$$\alpha_S(F) \cdot F = \frac{1}{L}\sum_{l=1}^{L}\sum_{k=1}^{K} w_{l,k} \cdot F(l; p_k + \Delta p_k; c) \cdot \Delta m_k, \tag{3}$$

where $K$ represents the number of sparse sampling positions, $p_k + \Delta p_k$ represents positions that can be moved through self-learned spatial offset $\Delta p_k$, and $\Delta p_k$ is learned *via* deformable convolution to focus on a specific region of interest. $\Delta m_k$ represents the importance of self-learned positions $p_k$. By introducing the spatial-aware attention module, the features become sparser, thus more effectively focusing on tomato leaf targets at different spatial positions.

**Task-aware attention** $\alpha_C(F)$. Due to variations in factors such as tomato leaf morphology and lighting conditions, it is imperative for the model to possess strong robustness. The task-aware attention can dynamically switch features from different channels, thereby effectively enhancing the robustness of detection:

$$\alpha_C(F) \cdot F = max[\alpha^1(F) \cdot F_c + \beta^1(F), \alpha^2(F) \cdot F_c + \beta^2(F)], \tag{4}$$

where $max(\cdot)$ represents a function for activating thresholds in different channels, and $F_c$ denotes the features of the c-th channel. Specifically, as shown in Fig. 2, the process begins by utilizing average pooling to reduce dimensions, followed by two linear layers and normalization to obtain the final features.

Note that the enhanced module DyHead can be stacked multiple times, enabling the model to have stronger representation ability, effectively improving identification performance. The impact of varying numbers of DyHead modules on model performance is discussed in "Experiment".

### Hard target identification of tomato disease spot by fusion focaler-SIoU

To address the challenges posed by irregularly shaped and difficult-to-classify tomato disease samples, this article introduces Focaler-SIoU, which effectively improves the accuracy of bounding box regression and emphasizes the recognition of hard samples.

YOLOv4 uses the CIoU (*Zheng et al., 2020*) for bounding box regression by default, which takes into account the similarity between the ground truth boxes and predicted boxes.

$$CIoU = IoU - \frac{\rho^2(b, b^{gt})}{c^2} - \beta v, \tag{5}$$

$$\beta = \frac{v}{(1 - IoU) + v}, \quad v = \frac{4}{\pi^2}\left(\arctan\frac{w^{gt}}{h^{gt}} - \arctan\frac{w}{h}\right)^2, \tag{6}$$

where $w^{gt}$, $h^{gt}$, $w$, and $h$ represent the width and height of the ground truth bounding box and the predicted bounding box, respectively. However, the CIoU does not consider the impact of the angles between bounding boxes. Thus, we introduce SIoU (*Gevorgyan, 2022*):

$$SIoU = IoU - \frac{(\Delta + \Omega)}{2}. \tag{7}$$

This further considers the angular compatibility between ground truth boxes and predicted boxes, thereby effectively enhancing the accuracy of bounding box regression.

$$\Lambda = 1 - 2 \cdot \sin^2\left(\arcsin(z) - \frac{\pi}{4}\right), \tag{8}$$

$$z = \frac{c^h}{d} = \sin(\varphi), \quad c^h = \max(b_{cy}^{gt}, b_{cy}) - \min(b_{cy}^{gt}, b_{cy}), d = \sqrt{(b_{cx}^{gt} - b_{cx})^2 + (b_{cy}^{gt} - b_{cy})^2}, \tag{9}$$

where $\Lambda$ is the angle cost. $b_{cx}^{gt}$, $b_{cy}^{gt}$, and $b_{cx}$, $b_{cy}$ are the coordinate values of the centers of the ground truth box and the predicted box, respectively. $c^h$ and $d$ denote the height displacement and Euclidean distance between the centers of the ground truth bounding box and the predicted bounding box. By minimizing $z$, the angle $\varphi$ between the ground truth box and the predicted box can be minimized.

$$\Delta = \sum_{t=x,y}(1 - e^{-\tau \rho_t}), \quad \tau = 2 - \Lambda, \ \rho_x = \left(\frac{b_{cx}^{gt} - b_{cx}}{c^w}\right)^2, \ \rho_y = \left(\frac{b_{cy}^{gt} - b_{cy}}{c^h}\right)^2, \tag{10}$$

where $\Delta$ is the distance cost, and $c^w$ denotes the width displacement between the centers of the ground truth bounding box and the predicted bounding box. By incorporating the angle cost into the distance cost, when $\varphi$ increases, the contribution of the distance cost also increases.

$$\Omega = \sum_{t=w,h}(1 - e^{-\Phi_t})^\theta, \quad \Phi_w = \frac{|w - w^{gt}|}{\max(w, w^{gt})}, \ \Phi_h = \frac{|h - h^{gt}|}{\max(h, h^{gt})}, \ \theta = 4, \tag{11}$$

where $\Omega$ is the shape cost. For different datasets, the value of $\theta$ varies. The variation of $\theta$ affects the rate of exponential decay, influencing the extent to which $\Omega$ penalizes differences between the ground truth box and the predicted box.

The obtained SIoU loss can be expressed as:

$$L_{SIoU} = 1 - IoU + \frac{(\Delta + \Omega)}{2}. \tag{12}$$

Due to the diversity of tomato leaf diseases and the differences in shape, color, and texture between individuals, there is a significant proportion of hard samples encountered during the recognition process. We introduced Focaler-IoU, which focuses more on the bounding box regression of hard samples. Specifically, we reconstruct the IoU loss through linear interval mapping, aiming to perform more accurate bounding box regression.

$$FocalerIoU = \begin{cases} 0, & \text{if } IoU < k \\ \dfrac{IoU - k}{g - k}, & \text{if } k \ll IoU \ll g, \quad \text{where } [k, g] \in [0, 1]. \\ 1, & \text{if } IoU > g \end{cases} \tag{13}$$

By varying the factors $g$ and $k$, we can increase the emphasis on the regression of hard samples. We set $k = 0$ and $g = 0.95$ in this article. The final Focaler-SIoU loss can be expressed as:

$$L_{Focaler-SIoU} = L_{SIoU} + IoU - FocalerIoU. \tag{14}$$

By incorporating the Focaler-SIoU loss, the proposed model focuses more on hard samples and improves the detection capability for small targets associated with tomato disease spots. This further enhances the model's learning ability for hard samples.

### Base model

YOLOv4 (tiny) is selected as the baseline model due to its optimized balance between speed and accuracy, making it ideal for real-time applications. The model utilizes a CSPDarknet53-Tiny backbone network to extract global features from 416,416 pixel tomato disease images efficiently. These features are crucial for both disease classification and detection tasks.

The network outputs two sets of feature layers at resolutions of 1,313 and 2,626, which are utilized for classification and detection purposes. It performs classification to identify the type of tomato disease present in the image and simultaneously conducts object detection to pinpoint the location and size of each detected disease instance.

Overall, YOLOv4 (tiny) excels in efficiently processing tomato disease images, providing detailed outputs that include disease classification, precise localization, and confidence scores for each detected instance. This capability is essential for applications requiring swift and accurate assessment of plant health in agricultural settings.

## EXPERIMENT

### Experiment setting and evaluation metric

Before training each network model, all images are resized uniformly to $416 \times 416$ pixels. The batch size is set to 16, with a learning rate of 0.001 and weight decay of 0.94. Training employs the Adam optimizer across 200 epochs, with model checkpoints saved after each epoch. Python serves as the primary programming language for both training and inference, leveraging hardware comprising an NVIDIA GeForce GTX 1080 Ti GPU and Intel Core i7 CPU.

In this article, mean average precision (MAP), average detection speed, average precision (AP), and F1 score are chosen as evaluation metrics to assess the identification performance of various models for tomato diseases. The specific calculation methods are as follows:

$$MAP = \frac{1}{C}\sum AP_i, \tag{15}$$

$$AP = \int_0^1 P(R)dR, \tag{16}$$

$$F1 = \frac{2 \cdot \text{precision} \cdot \text{recall}}{\text{precision} + \text{recall}}, \tag{17}$$

$$\text{Precision} = \frac{TP}{TP + FP}, \tag{18}$$

$$\text{Recall} = \frac{TP}{TP + FN}, \tag{19}$$

where $C$ is the number of tomato disease categories and $AP_i$ is the AP value of each tomato disease category. AP and F1 consider the evaluation values of precision and recall. True Positive (TP) indicates that the category information of tomato leaf disease is detected and classified successfully. True Negative (TN) indicates that the category information of the tomato leaf disease image is detected but classified incorrectly. False Negative (FN) indicates that the category information of tomato leaf disease is not detected and classified incorrectly. False Positive (FP) indicates that the category information of tomato leaf disease is not detected but classified correctly.

## Data preparation

In this study, tomato leaf images were sourced from the publicly available dataset "Leaf Type Detection" (*Make, 2023*). This dataset comprises images of various tomato leaf diseases and healthy tomato leaves, including tomato mold (86 photos), tomato leaf fine plaque (101 photos), tomato leaf spot (137 photos), tomato leaf flavivirus (68 photos), tomato early blight (77 photos), tomato mosaic virus (44 photos), tomato late blight (99 photos), and healthy tomato leaves (56 photos), totaling 668 images.

However, deep learning models require substantial data for effective training, and the original dataset had a limited number of samples, posing challenges for successful model training. Therefore, prior to model training, each image underwent random data augmentation to generate a total of 6,012 tomato leaf image samples. Data augmentation is crucial in plant leaf pathology detection models as it introduces variations in lighting conditions and rotational angles, simulating diverse photographic environments and conditions. This approach not only increases the diversity of the training data but also enhances the model's adaptability to different lighting and viewpoint conditions. Moreover, it strengthens the model's ability to generalize, reducing dependency on specific image features and thereby improving accuracy and robustness in practical leaf pathology identification applications.

Figure 3 illustrates examples of data augmentation applied to leaf pathology images, specifically showcasing variations in light intensity. Figure 3A displays a subset of tomato

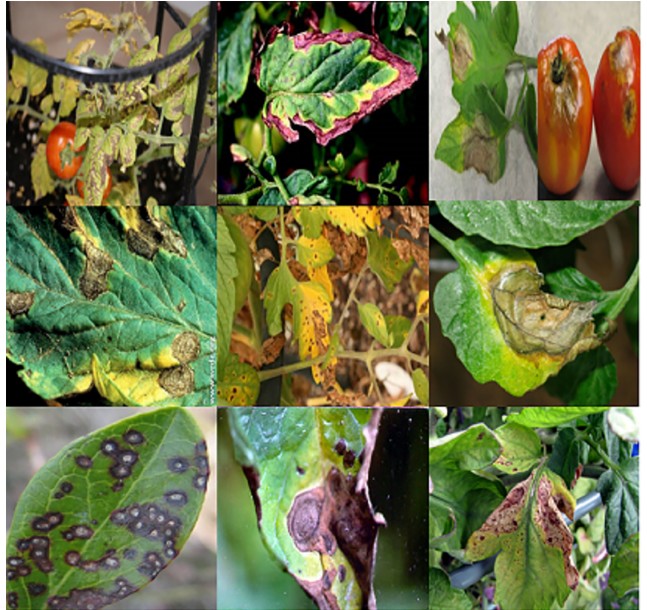
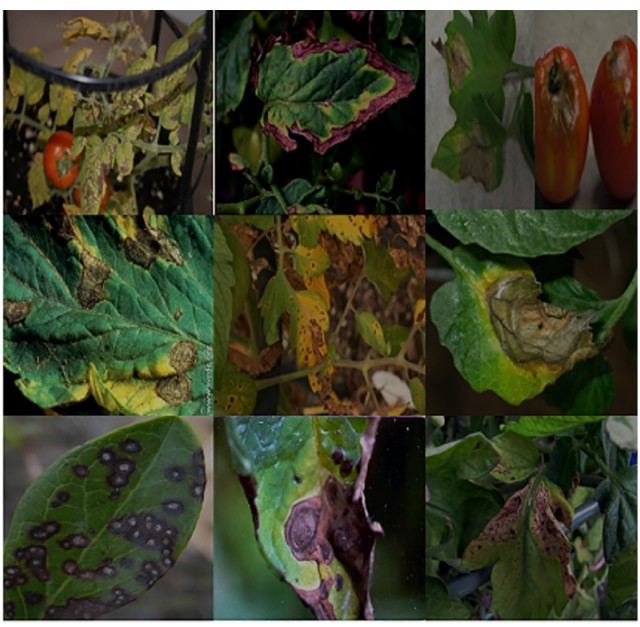

(a) Part of the data set before data enhancement    (b) Part of the data set after data enhancement

**Figure 3  Comparison of tomato leaf disease images before and after data enhancement.** Image source credit: PlantDoc dataset.

leaf disease images from the original dataset before augmentation, while Fig. 3B shows the same subset after augmentation. The dataset was split into training and test sets in a 9:1 ratio.

## Performance comparison

Table 2 shows the performance comparison of the enhanced module DyHead with different stacked numbers and various embedding dimensions. "nums" denotes the stacked number and "dims" denotes the embedding dimension. "speed" indicates the average time to detect an image.

## Ablation study

### The influence of the enhanced module DyHead on recognition performance

In subsequent experiments in this article, the base model YOLOv4 (tiny) without the enhanced module DyHead serves as the baseline. The purpose is to compare the performance of the base model with the performance of the models that incorporate different numbers of DyHead modules. DyHead modules with different numbers and embedding dimensions are introduced into the base model to analyze their impact on effective feature extraction and recognition performance of tomato diseases.

As shown in Table 2, we observe that the accuracy of detection increases continuously as the number of stacked DyHead modules and embedding dimensions rise. However, this enhancement comes with an increase in parameters and consequently, detection time. Considering both detection accuracy and speed, we ultimately opt to stack 2 DyHead

**Table 1 Performance comparison of the enhanced module DyHead with different stacked numbers and various embedding dimensions.** "nums" denotes the stacked number and "dims" denotes the embedding dimension. "speed" indicates the average time to detect an image.

| Model | Nums | Dims | mAP (%) | Speed (s) |
|---|---|---|---|---|
| YOLOv4 (baseline) | 0 | – | 83.31 | 0.0073 |
| YOLOv4 | 1 | 64 | 90.77 | 0.0098 |
| YOLOv4 | 2 | 64 | 92.00 | 0.0115 |
| YOLOv4 | 3 | 64 | 92.85 | 0.0137 |
| YOLOv4 | 4 | 64 | 93.13 | 0.0160 |
| YOLOv4 | 1 | 128 | 91.69 | 0.0083 |
| **YOLOv4** | 2 | 128 | 93.17 | 0.0110 |

**Table 2 Add performance comparison of different types of IoU.** "Focaler-IoU" indicates whether Focaler-IoU is used.

| Models | Type | Focaler IoU | mAP(%) | Speed (s) |
|---|---|---|---|---|
| YOLOv4 with DyHead | CIoU | ✗ | 93.17 | 0.0110 |
| YOLOv4 with DyHead | CIoU | ✓ | 93.15 | 0.0110 |
| YOLOv4 with DyHead | SIoU | ✗ | 93.27 | 0.0108 |
| **YOLOv4 with DyHead** | SIoU | ✓ | 93.64 | 0.0110 |

modules in the base model and set the embedding dimension to 128. The model achieves a detection accuracy of 93.17%, with a slight reduction in speed.

Hence, by introducing an appropriate number of DyHead modules, which combine multi-head attention mechanisms, it becomes possible to effectively extract relevant spatial and scale features, thereby enhancing the model's feature learning and representation capabilities. This proves crucial in achieving optimal model performance. For all subsequent experiments within this article, we employ 2 DyHead modules with embedding dimensions set to 128.

### The effect of Focaler-SIoU loss on recognition performance

The incorporation of Focaler-SIoU into the proposed algorithm represents a notable enhancement in recognition accuracy. As demonstrated in Table 1, the integration of Focaler-SIoU elevates the average detection accuracy of the method to 93.64%. This improvement, amounting to an increase of nearly 0.5% compared to the variant without Focaler-SIoU, is achieved without compromising the algorithm's average detection speed.

Focaler-SIoU allows the algorithm to effectively train on scenes featuring a higher prevalence of challenging instances of tomato diseases. These "hard samples" typically involve diseases that are more nuanced or less frequently encountered, posing greater difficulty for traditional detection models. By prioritizing these challenging cases through the Focaler-SIoU mechanism, the algorithm can allocate more resources and attention during training, thereby refining its recognition capabilities specifically for these scenarios.

**Table 3 Model performance comparison.** The bolded values in the mAP and Speed columns represent the best-performing results in the experiment.

| Models | mAP (%) | Speed (s) |
|---|---|---|
| YOLOv4 (baseline) | 83.31 | **0.0073** |
| FasterRCNN (*Ren et al., 2015*) | 88.26 | 0.0373 |
| SSD (*Liu et al., 2016*) | 88.78 | 0.0138 |
| YOLOv3 (*Redmon & Farhadi, 2018*) | 90.09 | 0.0152 |
| YOLOv7 (Tiny) (*Wang, Bochkovskiy & Liao, 2023*) | 91.29 | 0.0114 |
| YOLOv5 | 92.44 | 0.0135 |
| **Ours** | **93.64** | 0.0110 |

**Table 4 Comparison of AP indexes of each type of tomato samples in different models.**

| Models | AP/% (IoU = 0.5) | | | | | | | |
|---|---|---|---|---|---|---|---|---|
| | Leaf mold | Bacterial spot | Septoria leaf spot | Tomato yellow leaf curl virus | Early blight | Tomato mosaic virus | Late blight | Healthy |
| YOLOv4 (baseline) | 85.57 | 83.80 | 94.89 | 47.31 | 86.23 | 84.21 | 97.53 | 86.91 |
| **Ours** | **95.26** | **96.91** | **99.41** | **65.23** | **98.01** | **96.89** | **99.87** | **97.55** |

**Table 5 Comparison of F1 indexes of each type of tomato samples in different models.**

| Models | F1 (IoU = 0.5) | | | | | | | |
|---|---|---|---|---|---|---|---|---|
| | Leaf mold | Bacterial spot | Septoria leaf spot | Tomato yellow leaf curl virus | Early blight | Tomato mosaic virus | Late blight | Healthy |
| YOLOv4 (baseline) | 0.78 | 0.78 | 0.89 | 0.45 | 0.79 | 0.80 | 0.93 | 0.79 |
| **Ours** | **0.91** | **0.91** | **0.96** | **0.63** | **0.93** | **0.94** | **0.99** | **0.94** |

The observed increase in accuracy underscores the efficacy of Focaler-SIoU in bolstering the algorithm's performance in tomato disease recognition tasks. Focaler-SIoU not only improves overall detection rates but also enhances the algorithm's robustness in handling diverse and challenging conditions commonly encountered in agricultural field settings.

### Compared with the base model

As shown in Table 3, the experimental results demonstrate that our proposed algorithm effectively improves the recognition accuracy of tomato diseases while maintaining a balanced recognition speed. Our model achieves 93.64% accuracy, which is a 10.3% improvement compared to the YOLOv4 (tiny)/baseline model. It even surpasses YOLOv7 (tiny) at 91.29% and YOLOv5 at 92.44%, with faster detection speed.

More specifically, we compared our proposed model with the YOLOv4 (tiny)/baseline model in terms of AP and F1 score for each type of tomato disease, as shown in Tables 4

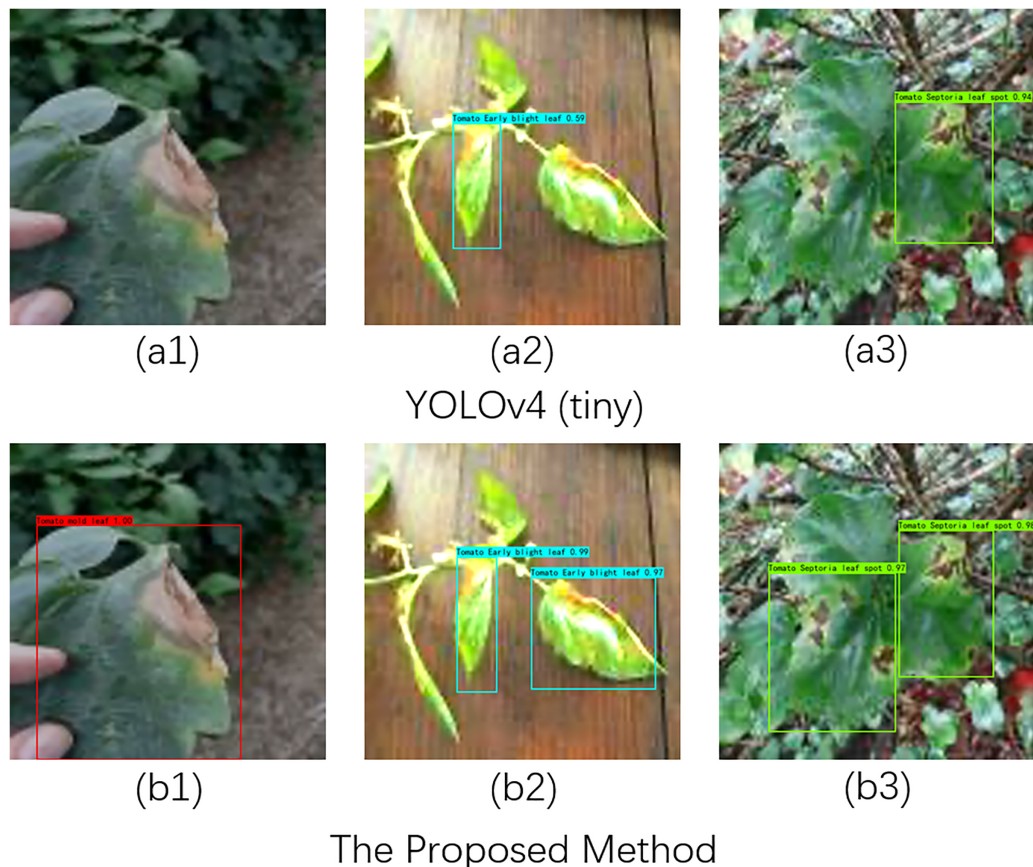

**Figure 4 Recognition effects of different network models on tomato leaf images under illumination changes.** Image source credit: PlantDoc dataset.

and 5. Notably, for tomato leaf flavivirus, the proposed algorithm exhibits the highest increase in AP value, with an improvement of about 17.9% compared to the YOLOv4 (tiny)/baseline model. Additionally, for tomato leaf flavivirus and healthy tomato leaves, the proposed method achieves improvements of 18% and 15% in F1 value, respectively, compared to the YOLOv4 (tiny)/baseline model.

These results validate that our proposed method effectively completes feature extraction for each type of tomato disease, accurately identifies hard tomato samples, and improves the overall recognition effect of the algorithm for each type of tomato disease. The experimental findings confirm the effectiveness of the proposed method in enhancing the recognition accuracy of tomato diseases.

## The influence of illumination variation on recognition and detection performance

In practical scenarios, detecting tomato leaf diseases is often challenged by complex backgrounds and varying lighting conditions. To evaluate the robustness of our proposed method for recognizing and detecting tomato diseases under different lighting environments, we selected three types of tomato diseases—tomato leaf mold, tomato leaf spot, and tomato leaf early blight—as our test subjects. We established three experimental

setups, as illustrated in Fig. 4 (A1–B1), Fig. 4 (A2–B2), and Fig. 4 (A3–B3), to conduct tomato disease recognition experiments under low light, normal light, and bright light conditions.

Specifically, the comparison between Fig. A1 and Fig. B1 indicates that YOLOv4 (tiny) failed to recognize tomato diseases under low light conditions, whereas our proposed method accurately detected the lesions. In normal light and complex background conditions, as shown in Fig. A2 and Fig. B2, our method identified two lesions, while YOLOv4 (tiny) detected only one. In the comparison between Fig. A3 and Fig. B3, it is evident that YOLOv4 (tiny) missed detecting one leaf lesion under strong light conditions. In summary, these experiments demonstrate that our proposed method exhibits robust performance in detecting tomato diseases across different lighting conditions.

## CONCLUSIONS

In this article, we proposed a real-time tomato disease recognition algorithm using multi-head attention enhancement to address challenges in both recognition accuracy and speed for tomato leaf diseases. Our algorithm is based on YOLOv4 (tiny) and integrates a multi-head attention mechanism to accurately extract key features from complex backgrounds within tomato disease regions. Additionally, to further enhance the model's ability to classify different lesions at a fine-grained level, we incorporate the Focaler-SIoU method to handle classification samples with varying levels of difficulty. We conducted extensive experiments to demonstrate that the proposed method not only significantly improves detection accuracy under complex backgrounds and varying lighting conditions but also maintains a high detection speed, thus facilitating the application of the detection model in real-world scenarios.

In future work, we aim to enhance the algorithm's adaptability to diverse agricultural scenarios, target variations, and noise disturbances. This will involve diversifying the dataset, expanding the algorithm's capabilities to recognize different types of crop diseases, pests, and abnormalities, and developing techniques to handle noise disturbances commonly found in agricultural environments. By focusing on these aspects, we strive to create a more robust and versatile crop disease recognition algorithm that contributes to improved crop management and disease prevention in various agricultural settings.

### Funding
This work was supported by the Project of Shanxi Provincial Department of Education (No. 21JK0868). The funders had no role in study design, data collection and analysis, decision to publish, or preparation of the manuscript.

### Grant Disclosures
The following grant information was disclosed by the authors:
Shanxi Provincial Department of Education: 21JK0868.

## Competing Interests

The authors declare that they have no competing interests.

## Author Contributions

- Yumeng Yao conceived and designed the experiments, performed the computation work, prepared figures and/or tables, and approved the final draft.
- Xiaodun Deng conceived and designed the experiments, prepared figures and/or tables, and approved the final draft.
- Xu Zhang analyzed the data, authored or reviewed drafts of the article, and approved the final draft.
- Junming Li performed the experiments, prepared figures and/or tables, and approved the final draft.
- Wenxuan Sun performed the experiments, analyzed the data, authored or reviewed drafts of the article, and approved the final draft.
- Gechao Zhang conceived and designed the experiments, performed the computation work, authored or reviewed drafts of the article, and approved the final draft.

## Data Availability

The code and processed data are available on Figshare: Zhang, Xu (2024). code of "Automatic visual recognition for leaf disease based on enhanced attention mechanism". figshare. Software. https://doi.org/10.6084/m9.figshare.27210138.v1.

The original PlantDoc dataset is available at GitHub: https://github.com/pratikkayal/PlantDoc-Object-Detection-Dataset/tree/master.

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
