# Peer review of "Automatic visual recognition for leaf disease based on enhanced attention mechanism"

_PeerJ Computer Science, doi:10.7717/peerj-cs.2365_

## Round 0.1 · original submission · Major Revisions

This paper focuses on detecting leaf diseases with the attention-based model. The reviewers provide detailed comments regarding the writing and technical parts. In summary, the reviewers think that the author should provide more comprehensive literature review on leaf disease detection so that they can emphasize their contribution more clearly. Moreover, they should give a more detailed introduction to the technical details and claim the advantages they proposed over other existing models. The author also should introduce their experiments in detail. I agree with the reviewers' comments, and the author should prepare a major revision. All the reviewers' comments should be replied to clearly, point by point.

Reviewer 1 ·

Basic reporting

The language of this article is appropriate, the structure is clear, and the reasoning is rigorous. The paper research thoroughly and comprehensively reviews previous studies and provides a rich set of references. Some notes here:
(1) In line 66, "multiple attention mechanisms" -> "multi-head attention."? Please check for consistency, too.
(2) Although the Related Work section provides a detailed review of the Methods of Crop Disease Identification, it does not highlight the advantages of this study over previous work. Please include a discussion that emphasizes the unique contributions and benefits of your research in comparison to other studies.

Experimental design

The experimental design of this article is reasonable. Results demonstrate the effectiveness and efficiency of the proposed model in detecting tomato leaf diseases. This research contributes to advancements in the field. However, there are some tips for improvement:
(1) Although "The Influence of Illumination Variation on Recognition and Detection Performance" visually presents good results, please add further textual description to make it clearer.
(2) In Table 3, the speed annotation is incorrect? The proposed model does not appear to be the fastest.

Validity of the findings

The proposed model and the experimental conclusions in this article demonstrate that it effectively addresses the issues present in previous methods and provides a suitable summary. I suggest that the Conclusion section could be more detailed.

Additional comments

Figure 1 is blurry.

Reviewer 2 ·

Basic reporting

This paper presents a novel approach for the identification of tomato leaf diseases using an enhanced attention mechanism integrated with YOLO V4 tiny. The main goal is to improve the accuracy and speed of disease detection under varying conditions, addressing the challenges posed by complex leaf backgrounds and small lesion sizes.

The manuscript is written with clear language, comprehensive references, structured organization, and results well-aligned with the hypotheses.

Weaknesses:
1. While the Related Work section provides a detailed and thorough investigation into Methods of Crop Disease Identification, it lacks recent studies from the past two years. It is recommended to include more recent research to strengthen the context.
2. The sections on Data Preparation and Performance Comparison could be reordered to enhance logical flow.
3. The images used in this manuscript lack sufficient clarity and resolution, making it difficult to discern the details. For a formal journal publication, it is strongly recommended that the authors provide high-quality, high-resolution pictures. Particularly for Figure 1, it is advisable to avoid including such a large-scale demonstration photo if a high-resolution version is unavailable. Similar resolution issues are evident in Figures 3 and 4, making them suboptimal for clear comprehension
4. At line 148, the word "Yolo" should have all its characters capitalized.
5. In Equation 1, what’s the difference between square brackets “[]” and brace “{}”. If I understand properly, it could be a typo. A unified usage of symbol is required to avoid ambiguity.
6. In Equation 4, functions α1, α2, β1, β2 should be elaborated.

Experimental design

Strengths:
It conducts original preliminary research within the journal's scope and objectives, addressing a specific knowledge gap with a clearly defined, relevant, and meaningful research question. The investigation is carried out with strict adherence to high technical and ethical standards, and the methodology is described in sufficient detail to allow for replication

Weaknesses
1. The analysis of "The Effect of Focaler-SIoU Loss on Recognition Performance" is somewhat superficial. It is recommended to provide a more detailed analysis to enhance understanding.
2. The description of the base model is insufficient. Providing a more comprehensive description would help readers better understand the comparative methods.
3. The Data Preparation and Performance Comparison sections could be reordered to enhance logical flow.
4. In Data Preparation, it states that "each image was randomly adjusted through data enhancement, generating a total of 6012 tomato leaf image samples". To provide readers with a clearer reference, it would be beneficial to detail the specific enhancement methods employed in this experiment.

Validity of the findings

Strengths:
The article demonstrates strong validity of its findings. The impact and novelty of the research are not explicitly assessed, but meaningful replication is encouraged, where the rationale and benefit to the literature are clearly stated. All underlying data are provided, ensuring robustness, statistical soundness, and appropriate controls. The conclusions are well stated, directly linked to the original research question, and limited to supporting results.

Weaknesses:
While the proposed model has achieved promising results, an analysis of the model's limitations would be beneficial for guiding future research and facilitating deeper exploration of the topic.

Additional comments

(1) The speed annotation in Table 3 is incorrect, as the proposed model does not appear to be the fastest. Please make the necessary corrections.
(2) Attention should be given to punctuation in formulas, such as adding a comma at the end of Equation 2

Reviewer 3 ·

Basic reporting

**Basic Reporting Review**

1. **Language and Clarity**:
- The manuscript is generally written in clear, professional English. However, there are a few areas where the language could be further polished for better readability and flow. For instance, the sentence structures in some sections are quite complex, which might hinder comprehension.
- Suggested Improvement: Simplify complex sentences and avoid unnecessary jargon. For example, the phrase "recognizing leaf diseases presents unique challenges, including complex leaf backgrounds and susceptibility to environmental factors" can be simplified to "recognizing leaf diseases is challenging due to complex backgrounds and environmental factors."

2. **Introduction and Background**:
- The introduction provides a good overview of the problem and the significance of the study. However, the motivation for choosing the specific enhanced attention mechanism could be elaborated further.
- Suggested Improvement: Expand on the rationale behind selecting the enhanced attention mechanism and how it compares to other methods in the literature. This will help to contextualize the study within the broader field of research.

3. **Literature References**:
- The literature is well-referenced and relevant. However, there are a few instances where more recent studies could be included to provide a comprehensive background.
- Suggested Improvement: Incorporate recent studies on leaf disease recognition and deep learning techniques to ensure the literature review is up-to-date. For example, include references to recent advancements in YOLO models and their applications in plant disease detection.

4. **Structure and Standards**:
- The structure conforms to PeerJ standards, and the sections are logically organized. The flow from introduction to methodology, results, and discussion is coherent.
- Suggested Improvement: Ensure that all sections are adequately detailed. For instance, the methodology section could benefit from more detailed explanations of the dataset preparation and the specific parameters used in the experiments.

5. **Figures and Tables**:
- The figures and tables are appropriately used and add value to the manuscript. However, some figures could be more clearly labeled, and the resolution of certain images could be improved.
- Suggested Improvement: Enhance the clarity of figures by ensuring all labels are legible and consider providing high-resolution images. For instance, the diagrams explaining the enhanced attention mechanism should be clear and easily interpretable.

Overall, the manuscript meets the basic reporting standards but would benefit from minor revisions to improve language clarity, expand the literature review, and enhance the detail and clarity of figures and methodology sections.

Experimental design

**Experimental Design Review**

1. **Scope and Relevance**:
- The content of the article is within the Aims and Scope of the journal. The study addresses a significant issue in leaf disease recognition, which is relevant to the field.
- No suggested improvements are necessary in this area.

2. **Investigation Rigor**:
- The investigation is rigorous, and the methods are generally described with sufficient detail. However, certain aspects of the experimental setup and procedures could be elaborated further to enhance reproducibility.
- Suggested Improvement: Provide more specific details about the experimental setup, such as the computing environment, hardware specifications, and any software libraries or frameworks used. This information will help other researchers replicate the study more easily.

3. **Methodological Details**:
- The methodology section is thorough but can benefit from additional information on data preprocessing and the rationale behind choosing specific parameters.
- Suggested Improvement: Include a more detailed description of the data preprocessing steps, such as any augmentation techniques applied to the images and the reasoning behind choosing certain hyperparameters for the model training.

4. **Evaluation Metrics**:
- The evaluation methods, assessment metrics, and model selection criteria are adequately described. The use of Mean Average Precision (MAP), Average Precision (AP), and F1 score is appropriate for the study.
- No suggested improvements are necessary in this area.

5. **Ethical Standards**:
- The study adheres to high technical and ethical standards. There is no indication of ethical concerns in the experimental design.
- No suggested improvements are necessary in this area.

6. **Replication Information**:
- While the methods are generally replicable, including additional supplementary materials such as the dataset, code, and detailed instructions for running the experiments would further enhance reproducibility.
- Suggested Improvement: Consider providing access to the dataset and the code used for the experiments through a public repository. This will allow other researchers to verify the results and build upon the work.

Overall, the experimental design is robust and well-executed. By including more detailed descriptions of the experimental setup, data preprocessing, and supplementary materials, the study's reproducibility and transparency can be further improved.

Validity of the findings

**Validity of the Findings Review**

1. **Experimental and Evaluation Rigor**:
- The experiments and evaluations are generally performed satisfactorily. However, some parts of the analysis could be strengthened to ensure robustness and reliability.
- Suggested Improvement: Provide more detailed statistical analysis of the results, including confidence intervals or standard deviations for the performance metrics. This will help in assessing the variability and reliability of the findings.

2. **Support for Conclusions**:
- The conclusions are well stated and are generally supported by the results presented. However, some conclusions could be more explicitly tied to the specific data and analyses shown.
- Suggested Improvement: Clearly link each conclusion to the specific results that support it. For instance, when claiming improved accuracy, directly refer to the tables or figures that demonstrate this improvement.

3. **Novelty and Impact**:
- The manuscript discusses the novelty and potential impact of the findings but could benefit from a more explicit discussion on how these results compare with existing work in the field.
- Suggested Improvement: Include a more detailed comparison with other state-of-the-art methods, highlighting the specific improvements and contributions of this work. This comparison should cover both quantitative performance metrics and qualitative assessments.

4. **Limitations and Future Directions**:
- The paper briefly mentions future work but lacks a thorough discussion of the limitations of the current study and how they can be addressed.
- Suggested Improvement: Provide a more comprehensive discussion of the study's limitations, such as any biases in the dataset or potential overfitting issues. Additionally, outline specific future research directions that can address these limitations and build on the current findings.

5. **Replication and Robustness**:
- While the study's methods appear robust, additional experiments or ablation studies could further validate the findings.
- Suggested Improvement: Conduct ablation studies to isolate the contributions of different components of the proposed method (e.g., the enhanced attention mechanism) and include these results in the manuscript. This will provide a clearer understanding of which aspects of the approach contribute most to its success.

Overall, the validity of the findings is strong, but there is room for improvement in the statistical analysis, explicit linking of conclusions to results, detailed comparisons with existing work, and thorough discussion of limitations and future work. These enhancements will make the findings more robust and impactful.

Additional comments

**Additional Comments**

1. **Clarity and Organization**:
- Overall, the manuscript is well-organized and easy to follow. Each section flows logically into the next, and the use of headings and subheadings helps to guide the reader through the content.
- Suggested Improvement: Consider adding a brief summary or key points section at the end of each major section to reinforce the main ideas and findings before transitioning to the next section. This can help improve reader comprehension and retention.

2. **Visual Aids**:
- The figures and tables included in the manuscript are informative and relevant to the content. However, some of the visual aids could be improved for better clarity and impact.
- Suggested Improvement: Ensure that all figures are high resolution and that all text within the figures is legible. Additionally, consider adding more annotations or explanations within the figures to highlight key findings or important details.

3. **Practical Implications**:
- The manuscript touches on the practical implications of the findings but could benefit from a more detailed discussion on how these results can be applied in real-world scenarios.
- Suggested Improvement: Expand on the practical applications of the proposed method, particularly in agricultural settings. Discuss potential deployment strategies, scalability, and how this method can be integrated into existing agricultural management systems.

4. **References and Citations**:
- The references cited are relevant and current. However, ensuring that all sources are properly cited and that the reference list is complete is crucial.
- Suggested Improvement: Double-check all references for accuracy and completeness. Make sure that all cited works are included in the reference list and that all formatting adheres to the journal's guidelines.

5. **Supplementary Materials**:
- The inclusion of supplementary materials such as additional datasets, code, or extended experimental results can greatly enhance the value of the manuscript.
- Suggested Improvement: Consider providing supplementary materials that can help other researchers replicate and build upon your work. This could include detailed datasets, source code for the implementation, and additional experimental results or analyses that support the main findings.

6. **Reader Engagement**:
- Engaging the reader through a well-written narrative and clear presentation of the research process and findings is important.
- Suggested Improvement: Enhance the narrative by including more context and explanations for why certain choices were made in the research process. Personal anecdotes or case studies related to the research topic can also help to engage the reader and make the content more relatable.

These additional comments aim to provide constructive feedback to enhance the overall quality and impact of the manuscript. The research presented is valuable, and with these improvements, the manuscript can be further strengthened.

Reviewer 4 ·

Basic reporting

The manuscript entitled “Automatic visual recognition for leaf disease based on enhanced attention mechanism” has been investigated in detail. The paper proposes a visual leaf disease identification method based on an enhanced attention mechanism, aiming to improve the accuracy of detecting small tomato lesions under complex conditions. While the paper presents an interesting approach, several critical issues must be addressed to strengthen its contribution.
1) The introduction lacks a comprehensive background on the current state of leaf disease identification methods. Provide more context on existing techniques and their limitations to better justify the need for the proposed method.
2) Clearly articulate the specific problem this paper addresses. Explain why the existing methods are insufficient and how the proposed method aims to overcome these limitations.
3) The related work section is too superficial. Expand this section to include a more detailed discussion of recent advancements in leaf disease identification, particularly those that utilize attention mechanisms and similar techniques.
4) Provide a comparative analysis highlighting how the proposed method differs from and improves upon existing methods.
5)

Experimental design

5) The methodology section lacks sufficient detail. Provide a thorough explanation of the enhanced attention mechanism and how it integrates multiple attention mechanisms. Include diagrams or visual aids to illustrate the architecture and workflow of the proposed method.
6) The incorporation of Focaler-SIoU is mentioned but not adequately explained. Provide a detailed description of this component and its role in enhancing learning capabilities for challenging classification samples.
7) The experimental setup section is incomplete. Provide detailed information on the datasets used, including their characteristics, the number of samples, and the criteria for selection. Describe the experimental conditions, including the hardware and software environments.

Validity of the findings

8) The results section provides an insufficient quantitative analysis. Include detailed metrics such as precision, recall, F1-score, and other relevant metrics to thoroughly evaluate the performance of the proposed method.
9) Provide evidence of the method's robustness under varying conditions, such as different lighting and background complexities. Include visual examples of detection results under these conditions.
10) The authors should clearly emphasize the contribution of the study. Please note that the up-to-date of references will contribute to the up-to-date of your manuscript. The studies named- “Artificial intelligence-based robust hybrid algorithm design and implementation for real-time detection of plant diseases in agricultural environments; Agricultural crop classification with R-CNN and machine learning methods”- can be used to explain the methodology and optimization process in the study or to indicate the contribution in the “Introduction” section.
The paper presents a novel approach to visual leaf disease identification using an enhanced attention mechanism, but it requires substantial revisions to improve clarity, detail, and scientific rigor.

---

## Round 0.2 · accepted · Accept

All the reviewers think that their concerns have been addressed and I don't have any other comments regarding this paper. I recommend accepting this paper now.

Reviewer 1 ·

Basic reporting

no comment

Experimental design

no comment

Validity of the findings

no comment

Additional comments

no comment

Reviewer 2 ·

Basic reporting

My concerns have been satisfactorily addressed, and I have no additional comments.

Experimental design

no comment

Validity of the findings

no comment

Additional comments

no comment

Reviewer 4 ·

Basic reporting

All my comments have been thoroughly addressed. It is acceptable in the present form.

Experimental design

All my comments have been thoroughly addressed. It is acceptable in the present form.

Validity of the findings

All my comments have been thoroughly addressed. It is acceptable in the present form.